# Peer review of "Coeliac Disease in Elderly Patients: Value of Coeliac Lymphogram for Diagnosis"

_nutrients, 2021, doi:10.3390/nu13092984_

Round 1
Reviewer 1 Report
In this multicenter retrospective study conducted in Spain authors evaluated the sensitivity of intraepithelial lymphocyte cytometric patterns for celiac disease (CD) diagnosis in older patients (> 50 yrs). In CD patients older than 70, a high diagnostic sensitivity was found by lowering the TCRgammadelta+ count cut-off from 14% to 10%.
Authors do not explain the basis for choosing the 10% instead than the 14% cut-off, no ROC analysis was apparently done to justify this change. It looks like the new cut-off was chosen on the basis of the study-results, but these can change from one study-group to another. Furthermore decreasing the cut-off to maximize the sensitivity negatively influnces the specificity of the test, however this negative consequence and its impact on clinical practice is neither evaluated nor discussed at all.
Author Response
REVIEWER 1
In this multicenter retrospective study conducted in Spain authors evaluated the sensitivity of intraepithelial lymphocyte cytometric patterns for celiac disease (CD) diagnosis in older patients (> 50 yrs). In CD patients older than 70, a high diagnostic sensitivity was found by lowering the TCRgammadelta+ count cut-off from 14% to 10%.
Authors do not explain the basis for choosing the 10% instead than the 14% cut-off, no ROC analysis was apparently done to justify this change. It looks like the new cut-off was chosen on the basis of the study-results, but these can change from one study-group to another. Furthermore decreasing the cut-off to maximize the sensitivity negatively influences the specificity of the test, however this negative consequence and its impact on clinical practice is neither evaluated nor discussed at all.
Response: The cut-off for TCRgδ+ cells fluctuated among centres from 8.5 to 12%, in most of them was 10%, and was validated in each lab using the appropriate studies. These differences in the cut-off imply slight variations in sensitivity and specificity between centres, the lower the cut-off, the higher the sensitivity and the lower the specificity, and vice versa. Some centres consider specificity and others sensitivity to be more important. The meta-analysis of the published studies using these cut-offs between 8.5 and 12% showed a pooled sensitivity and specificity of 95%. Specifically using the same criteria of the present manuscript (a cut-off of TCRgδ+ >10% and of CD3- <10%), we showed a specificity of 97% in seronegative CD in a multicentre study (Fernández-Bañares et al. Aliment Pharmacol Ther 2020;51:699-705). At least a similar value should be expected for the whole CD.
A cut-off of 14% was that used in Nijeboer study.
We have clarified these points in the paper: lines 114 to 124.
Reviewer 2 Report
Reviewer's report:
Title: Coeliac disease in elderly patients: the value of coeliac lymphogram for diagnosis
Version: 1st version
Date: 26.07.2021
Reviewer's report:
The content of the subject, "Coeliac disease in elderly patients: the value of coeliac lymphogram for diagnosis", has a potential clinical value of interest; however, I believe it needs to be majorly revised before the consideration for submission.
- Major Revisions
- Line 98-103: Authors should present the ethical approval number of the study.
- Line 112: Authors mentioned that the results from different centres regarding the CD3- and TCRγδ+ cells were almost identical. To give their work more transparency, I think authors should present these results in tabular form with a statistical test as a supplementary file.
- Line 114-124: authors mention that because of the different sensitivity and specificity preferences, the centres have different cut-offs; however for this study, they took one common cut off for all. I believe then they should present the sensitivity and specificity of this common cut off.
- Line 127: From the information, I would then assume there were different analyses from different centres? If so, what were the ELISA tests?
- Line 140: Authors used mean ± SEM; however, the SEM is a measure of precision for an estimated population mean. Unlike SD, SEM is not a descriptive statistic and should not be used as such. The SEM is correctly used only to indicate the precision of the estimated mean of the population. Even then, however, a 95% confidence interval should be preferred. Thus, I recommend authors to use SD instead of SEM values.
- Table 1 need explanation under the table, the statistical test etc., like others.
- Table 2 and 3: p values should be presented in tables, not under the table.
- I would also be interested to know the specificity as the sensitivity alone is not sufficient to define the diagnostic accuracy of the test.
Author Response
REVIEWER 2
Line 98-103: Authors should present the ethical approval number of the study.
Response: The ethical approval data number has been added to the manuscript (line 308).
Line 112: Authors mentioned that the results from different centres regarding the CD3- and TCRγδ+ cells were almost identical. To give their work more transparency, I think authors should present these results in tabular form with a statistical test as a supplementary file.
Response: In all the participating centres, development and fine-tuning of the methodology to analyse intraepithelial lymphocytes by flow cytometry was performed following the indications of the pioneer lab of this methodology in Spain (Immunology Department of the Hospital Ramón y Cajal, Madrid, Spain). In all cases, several samples were analysed in parallel to confirm proper performance. In the Appendix 1, the concordance between different cytometres and between one of the centres and Hospital Ramón y Cajal is shown.
Line 114-124: authors mention that because of the different sensitivity and specificity preferences, the centres have different cut-offs; however for this study, they took one common cut off for all. I believe then they should present the sensitivity and specificity of this common cut off.
Response: see answer to reviewer 1. The meta-analysis of the published studies showed a pooled sensitivity and specificity of 95% and using a cut-off of TCRgδ+ >10% and of CD3- <10%, we showed a sensitivity of 87% and specificity of 97% in seronegative CD in a multicentre study.
Line 127: From the information, I would then assume there were different analyses from different centres? If so, what were the ELISA tests?
Response: As the reviewer points out, different commercially available quantitative ELISA kits were used in the different centres, which is usual in clinical practice. These kits are: Elia Celikey IgA (Thermo Fisher, Phadia AB, Uppsala, Sweden), Aeskulisa tTg-A (Aesku.Diagnostics, Wendelsheim, Germany) and Bioplex 220 Celiac IgA (BioRad, California, USA). EMA was used to confirm a positive result in all samples tested with the Aeskulisa tTg-A kit. In the remaining centres, EMA was usually tested in samples showing low anti-TG2 antibody titres. This information has been added to the manuscript (lines 133-140).
Line 140: Authors used mean ± SEM; however, the SEM is a measure of precision for an estimated population mean. Unlike SD, SEM is not a descriptive statistic and should not be used as such. The SEM is correctly used only to indicate the precision of the estimated mean of the population. Even then, however, a 95% confidence interval should be preferred. Thus, I recommend authors to use SD instead of SEM values.
Response: SD values have been included instead of SEM.
Table 1 need explanation under the table, the statistical test etc., like others.
Response: the information about the statistical methods is described in the statistics section. In the case of Table 1, results were compared with a Chi-square test since the variables were qualitative. Note that the information provided under Table 2 and 3 is not needed in Table 1.
Table 2 and 3: p values should be presented in tables, not under the table.
Response: In the Table there are asterisks when it was necessary describe the ‘p-value’ and for not complicate more the Table these values are described under it. Note that the described “p-values” do not correspond to specific rows.
I would also be interested to know the specificity as the sensitivity alone is not sufficient to define the diagnostic accuracy of the test.
Response: As mentioned, the specificity was not calculated in the present study. We have referenced recent papers calculating the specificity of the test [references 11-14]. The specificity was always very high regardless of age.
Reviewer 3 Report
The paper presents an important problem for the clinical practice: very late diagnosis of celiac disease in elderly. There is known that in this group of patients many discrepancies between intestinal biopsy, serology and intestinal lymphogram (IELs) occur.
The paper is overall interesting and of good quality, methodology and results presentation seem correct, however I have some questions and hints:
1) The reference for the definition of Marsh 1 lesions with IEL count >25 IELs per 100 epithelial nuclei should be added (line 137)
2) As far as I understand the materials and methods, among 127 patients included, were 6 patients having seronegative CD with villous atrophy. Generally, I am not sure was it correct to include these 6 seronegative patients to the 121 seropositive. What was the idea? In my opinion it is breaking the homogeneity of the study group. I understand that removing these 6 patients now, would destroy all the results, but would it be possible simple checking additionally the differences between these seronegative and seropositive patients or comment to that? Do the lymphogram in seronegative patients differ from seropositive? Was it assessed?
3) According to the current CD diagnosis criteria, positive double serology (TTG+EMA) and Marsh 2-3 in symptomatic patient should be regarded as final diagnosis and confirmed CD without any doubts at any age. Please make clear statement about it. The current version of discussion may suggest that TCR counting and proposed cut-off >10% is a part of diagnostic criteria of celiac disease. There should be clearly stated what are the diagnostic criteria of CD and for what purpose the intestinal lymphogram and TCR gamma/delta counting are made. I see it in discussion lines 249-259 (the importance of lymphogram in patients after long term GFD or in patients without diagnosis of CD due to GFD started before), but I suggest to change a little the discussion to have it more clearly stated.
Author Response
REVIEWER 3
The paper presents an important problem for the clinical practice: very late diagnosis of celiac disease in elderly. There is known that in this group of patients many discrepancies between intestinal biopsy, serology and intestinal lymphogram (IELs) occur.
The paper is overall interesting and of good quality, methodology and results presentation seem correct, however I have some questions and hints:
1) The reference for the definition of Marsh 1 lesions with IEL count >25 IELs per 100 epithelial nuclei should be added (line 137).
Response: It has been added (ref number 13).
2) As far as I understand the materials and methods, among 127 patients included, were 6 patients having seronegative CD with villous atrophy. Generally, I am not sure was it correct to include these 6 seronegative patients to the 121 seropositive. What was the idea? In my opinion it is breaking the homogeneity of the study group. I understand that removing these 6 patients now, would destroy all the results, but would it be possible simple checking additionally the differences between these seronegative and seropositive patients or comment to that? Do the lymphogram in seronegative patients differ from seropositive? Was it assessed?
Response: As mentioned in the material and methods section, seronegative Marsh 3 patients were diagnosed with CD based on the presence of both a coeliac pattern on IEL flow cytometry and a clinical and histological response to GFD [12,15]. This is a not uncommon situation in adult patients and we decide do not exclude them. In relation to the coeliac lymphogram, it was present, but the sample size precluded to perform statistical comparisons between seropositive and seronegative patients.
3) According to the current CD diagnosis criteria, positive double serology (TTG+EMA) and Marsh 2-3 in symptomatic patient should be regarded as final diagnosis and confirmed CD without any doubts at any age. Please make clear statement about it. The current version of discussion may suggest that TCR counting and proposed cut-off >10% is a part of diagnostic criteria of celiac disease. There should be clearly stated what are the diagnostic criteria of CD and for what purpose the intestinal lymphogram and TCR gamma/delta counting are made. I see it in discussion lines 249-259 (the importance of lymphogram in patients after long term GFD or in patients without diagnosis of CD due to GFD started before), but I suggest to change a little the discussion to have it more clearly stated
Response: We have added a sentence in the discussion (lines 258-261) to clarify these aspects.
Round 2
Reviewer 2 Report
Please add the statistical test information under the tables. Tables should be clearly understandable without checking the main text.
I still have my doubts about the sensitivity, specificity issue and the common cut-off however if the authors believe they provided enough information and transparency then I will respect their decision.
Author Response
REVIEWER 2
Please add the statistical test information under the tables. Tables should be clearly understandable without checking the main text.
Response: Statistical test information has been added under Tables 1 and 2, which contain statistical analyses.
I still have my doubts about the sensitivity, specificity issue and the common cut-off however if the authors believe they provided enough information and transparency then I will respect their decision.
Response: We consider enough information and transparency is provided. Different centres use different cut-offs depending on their main interest (higher sensitivity or higher specificity), but a cut-off of TCRgδ+ >10% and of CD3- <10%, as the one used in the manuscript, is the most balanced one and the one recommended to use before enough data have been collected in each centre and specific validations can be performed.